

# Comparison of gravity wave propagation direction observed by mesospheric airglow imaging at three different latitudes by using M-transform

Septi Perwitasari[1, 3], Takuji Nakamura[1, 2], Masaru Kogure[2, 1], Yoshihiro Tomikawa[1, 2], Mitsumu K. Ejiri[1, 2], Kazuo Shiokawa[4]

[1]National Institute of Polar Research, Tokyo, 190-8518, Japan

[2]Department of Polar Science, SOKENDAI (The Graduate University for Advanced Studies), Tokyo, 190-8518, Japan

[3]National Institute of Aeronautics and Space (LAPAN) of Indonesia, Bandung, 40173, Indonesia

[4]Institute for Space-Earth Environmental Research, Nagoya University, Nagoya, 464-8601, Japan

*Correspondence to*: Septi Perwitasari (septi.perwitasari@nipr.ac.jp)

**Abstract.** We have developed a user-friendly software based on *Matsuda et al*. (2014) 3D-FFT method (M-Transform) for airglow imaging data analysis, as a function on Interactive Data Language (IDL), in which users can customize the range of wave parameters to process when executing the program. Input of this function is 3D array of a time series of 2D airglow image in geographical coordinate. We have applied this new function to mesospheric airglow imaging data with slightly different observation parameters obtained for the period of April-May at three different latitudes; Syowa Station,

the Antarctic (69°S, 40°E), Shigaraki, Japan (35°N, 136°E), and Tomohon, Indonesia (1°N, 122°E). Day-to-day variation of phase velocity spectrum at Syowa Station was smaller and propagation direction was mainly westward. At Shigaraki, the day-to-day variation of horizontal propagation direction was larger than at Syowa Station, and the variation at Tomohon was even larger. At Tomohon variation of nightly power spectrum magnitude was remarkable, which suggests intermittency of atmospheric gravity waves (AGWs). The average of nightly spectrum in April – May showed that at

Syowa Station dominant propagation was westward with phase speed <50 m/s, and at Shigaraki east/south-eastward propagation with phase speed up to ~80 m/s was prevailing. A Tomohon, day-to-day variation was too strong to discuss about average characteristics, however it showed a phase speed up to ~100 m/s and faster.  The corresponding background wind profiles derived from MERRA-2 indicated that at Syowa Station, wind filtering played significant role in filtering out waves propagated eastward. On the other hand, at Shigaraki and Tomohon, the background winds were not strong

enough to filter out the relatively high speed AGWs and that the dominant propagation direction was likely more related to the distribution/characteristic of the source region, at least in April and May.



## 1 Introduction

Atmospheric gravity waves (AGWs) or buoyancy waves are oscillations caused by the vertical displacement of an air parcel, of which then restored to its initial position by buoyancy. These waves, whose origin is primarily in the lower atmosphere, propagate into the upper mesosphere and lower thermosphere (e.g., *Lindzen*, 1981; *Matsuno*, 1982;
*Fritts*, 1984; *Fritts and Alexander*, 2003). Study of AGWs is important because these waves transport energy and momentum vertically and drive general circulation up to ~100 km. AGWs can greatly affect global temperature structure through general circulation, and therefore correct understanding and implementation of the AGWs in numerical models, such as climate models, are very important to improve those models to have higher precision. However, quantitative observational studies are progressing slowly, partly because of the difficulties of investigating global characteristics of the
AGWs. One difficulty is that AGWs cover a very broad spectrum (wave periods range from minutes to hours and spatial scales extend up to thousands of km). Although satellite observations can provide a global view of AGWs phenomena, it is limited to only a portion of AGWs with a large scale because of low horizontal and/or vertical resolutions of the instruments (*Alexander,* 1998; *Wright et al.,* 2015). Therefore, study using ground-based measurements with high horizontal and/or vertical is essential to reveal global characteristics of AGWs.

Among various ground-based observations, airglow-imaging observation has been proven to be very effective to examine both energy and propagating characteristics. In the last few decades, airglow imagers have been deployed in various latitudes to study the AGWs in the equatorial, mid-latitudes and polar regions (e.g. *Shiokawa et al.*, 2009; *Suzuki et al.,* 2007; *Matsuda et al.*, 2014, 2017). Such long-term observations provide a huge amount of data. However, the lack of sophisticated analysis methods prevented quantitative studies using full amount of worldwide dataset that have been
collected with great efforts.

A new analysis algorithm for obtaining the power spectrum in the horizontal phase velocity domain from long series data of airglow has been developed in National Institute of Polar Research (NIPR) group as shown in *Matsuda et al.* (2014) (M-transform). The M-Transform method transforms airglow intensity image data to power spectrum in the horizontal phase velocity domain. This method can handle huge amount of data, and therefore it cuts time consumption
and manpower to analyze huge amount of airglow data. This method has been successfully applied to Antarctic Gravity Wave Instrument Network (ANGWIN) imagers to study the characteristics of mesospheric AGWs over Antarctica as reported by *Matsuda et al.* (2017). *Takeo et al.* (2017) and *Tsuchiya et al.* (2018) applied this method to 16 years airglow imaging data from Shigaraki (35°N, 136°E) and Rikubetsu (44°N, 144°E), Japan, to study the horizontal phase speed and propagation direction in the mid-latitude.

Despite the high success of this method's application, the original program needs to have several sub-routines executed separately, which is not user-friendly. Therefore, we have developed a simple and user-friendly function based on *Matsuda et al.* (2014) method that can be handled by users with different background of programming skill. We also aim to encourage AGWs research groups to use the program to analyse their data and produce result in a uniform format



(power spectrum in the phase velocity domain). In this manner, it would be easier to compare the AGWs phase velocity and energy distribution between different locations, i.e. latitudes and longitudes. This paper is organized as follows. Section 2 describes the M-transform function and how to use it. Section 3 shows comparison of phase speed spectra obtained by M-transform application on different airglow data set with different observation parameters taken from

different latitudes, Syowa Station (69°S, 40°E), Shigaraki (35°N, 136°E), and Tomohon Observatory (1°N, 122°E). The conclusion of this paper is given in Section 4.

## 2 M-Transform Function

Matsuda-transform (M-transform) function is a 3D FFT function on IDL (Interactive data Language,

*http://www.exelisvis.com/ProductsServices/IDL.aspx*) to analyze airglow imaging data, based on method developed by *Matsuda et al.* [2014]. This method requires time series of airglow images in 3D array format, which have been preprocessed to have fixed intervals of image pixel ($dx$, $dy$) and time ($dt$). These pre-processed images should be obtained through common airglow image pre-processing, i.e., (1) star removal, (2) correction of Van Rijhin effect, if necessary, and (3) projection into geographical coordinates in equal distances in latitudinal and longitudinal directions (e.g., *Kubota et al.*,

2001; *Suzuki et al.*, 2007). It is also recommended to normalize the airglow intensity values to compare the power spectral values from different dataset. For this purpose, $((I − \bar{I})/\bar{I})$ (where $I$ is the image intensity of each pixel) should be used in the input array. Our M-transform code applies 2D pre-whitening to the images to reduce the contamination of the lower wavenumber peaks of AGWs red spectrum to higher wavenumbers. It also applies the 2D Hanning window to reduce harmonics of the window function in horizontal directions (*Coble et al.*, 1998). *Matsuda et al.* (2014) new method

transforms the PSD (Power Spectrum Density) in wave number domain ($k$, $l$, $\omega$) to phase velocity domain ($v_x$, $v_y$, $\omega$) by using following equation:

$$v_x = \omega k/(k^2+l^2) \tag{1}$$
$$v_y = \omega l/(k^2+l^2) \tag{2}$$

Where $v_x$ and $v_y$ are the ortoghonal projection of the phase velocity to zonal and meridional axes, respectively (($v_x$, $v_y$) = $c$

($sin\ \varphi$, $cos\ \varphi$), where $c$ and $\varphi$ are phase speed and azimuth (east from north) of phase velocity). Note that $v_x$ and $v_y$ are not phase velocity along zonal or meridional directions or cross-sections. $\omega$ is the frequency; $k$ and $l$ are the zonal and meridional wavenumbers, respectively. Finally, the phase velocity spectrum is integrated over frequency and results in a 2D phase velocity spectrum.

The original program developed by *Matsuda et al.* (2014) consisted of several subroutines to calculate the 3D FFT,

making interpolation table, doing the interpolation and then plot the result separately. Users had to set the wave parameters in each subroutine and then had to run it one by one, making it complicated and inefficient. Furthermore, once the users failed to compile one of the subroutines, the program failed to run and the users had to check the subroutine one by one



again. Therefore, a simple and hassle-free with adjustable input keywords in one-line-command software is necessary to efficiently applying this method.

Table 1 shows the description of the M-transform function on IDL and Figure 1 shows a chart on how to run the program. In order to run this function, user can simply use the calling sequence *"Result=Matsuda_FFT(Img)"*, where "Img"

is 3D array of pre-processed images in geographic coordinate (*x*, *y*, *t*). The image resolutions (*dx*, *dy*, *dt*), wave parameters (horizontal wavelength (*LH*), wave period (*τ*), phase speed (*Vp*)) and size of zero padding (*zpx*, *zpy*, *zpt*) can be adjusted by setting the input parameters keywords. The default image resolutions are given as 1 km in both zonal (*dx*) and meridional (*dy*) direction. The default of image interval time (*dt*) is 60 s. The default minimum and maximum values for wave parameters are given as: $5 \leq LH \leq 100$ km, $8 \leq \tau \leq 60$ min, and $0 \leq Vp \leq 150$ m/s. The default output of this program is a 2D

phase velocity spectrum. In this new function, the default interpolation method is Delaunay triangulation method [e.g., *Dwyer*, 1987; *Su et al*., 1997]. One restriction of this program is that it requires equal time interval (*dt*). This may not be the case for practical observation, where filter rotations or background (or dark) images for calibration are obtained in the observation sequence. In that case, users have to interpolate the image to get constant *dt* interval before applying this function. The more details on this function development will be described elsewhere.

Figure 2 shows an example on how to use the M-transform function. The input was airglow data over Syowa Station on 20 September 2011 (the same data set as *Matsuda et al*., 2014), which had a dimension of [*x*=400 (km), *y*=400 (km), *t*=21 (x 3 min)]. Since the image time resolution was 3 minutes, the *dt* in input keyword was set as *dt*=180 s. The other wave parameters remained as default. The IDL console shows the input, calling sequence, total calculation time and output. The output can be seen as a 2D phase velocity spectrum plot in the right panel. The horizontal axis is the $v_x$, vertical axis is $v_y$

and the color bar shows the PSD in log scale. The phase velocity spectrum shows the dominant propagation in the southwestward direction, which agrees with the wave propagation seen in the airglow pre-processed image movie in supplement material S1 and as reported by *Matsuda et al*. (2014). The total calculation time of the M-transform function for 21 images with 400 x 400 pixels was ~1.4 min using a personal computer. This calculation was done on a MacBook Pro with a dual core 2.8 GHz Intel Core i7 processor with 4 MB cache size, and 16 GB memory, while the program was single

threaded.

**3 Comparison of phase velocity spectra obtained by M-transform function for different airglow data at three different latitudes**

We have applied this new M-transform function to various airglow data sets taken from three different latitudes.

Table 2 shows the summary of each airglow imager used in this analysis. Syowa Station (69°S, 40°E) data represents airglow data in polar region, while Shigaraki (35°N, 136°E) and Tomohon (1°N, 122°E) Observatory represent data in mid-latitude and equatorial region, respectively. We used Na (589.6 nm) airglow emission taken from Syowa Station and OI (557.7 nm) airglow emission taken from Shigaraki and Tomohon Observatory.





We observed 9 clear-sky nights at Syowa Station during the period of April-May 2013, 10 clear-sky nights at Shigaraki during April-May 2011 and 5 clear-sky nights at Tomohon during April-May 2016. Syowa data were the same data set that was reported by *Matsuda et al.* (2017). Image size for each observation site was 400x400 km and the sampling interval *(dt)* was 1 minute (Syowa), 5.5 minutes (Shigaraki) and 2.5 minutes (Tomohon). The difference of sampling interval

limits the maximum phase velocity to be analysed, and that at Shigaraki was the smallest. In order to avoid an aliasing effect on the 2D phase velocity spectrum, we had to set the period minimum ($T\_min$) at least 2 times larger than the sampling interval (2 x *dt*). In this study we set $T\_min$ as 3 x *dt*. Therefore, since Shigaraki data's *dt* was 5.5 minutes we set the $T\_min$ as 16.5 minutes for all the data sets in order to have the same frequency range. This setting can be easily done by changing the input keyword ($T\_min$) when running the function. The other wave parameter values were set as default.

Figure 3 shows the day-to-day variation of 2D phase velocity spectrum results at (a) Syowa Station (b) Shigaraki and (c) Tomohon Observatory. Topside is to the north and leftside is to the east. The day-to-day plot at Syowa Station shows lack of eastward GW propagation with a dominant westward propagation, probably due to the filtering effect by eastward polar night jet as described in *Matsuda et al.* (2014). However, the azimuth direction with large power spectral density was not always the same, especially on May 13 and 14, which show much narrower distribution in azimuth. This

result suggests that the distribution of the gravity wave sources were limited on these days, and changed day to day. On the contrary, 2D phase velocity spectrum at Shigaraki shows lack of westward propagation with a dominant eastward and southward propagation. The Shigaraki data showed much larger day to day variation in azimuth of strong spectrum than Syowa, suggesting that the distribution of the wave source region was more variable at Shigaraki at least in this season of April and May. At Tomohon, the day-to-day variation of propagation direction was even stronger. At the same time, it shows

significant day-to-day variation of magnitude of the power spectrum. The power was the weakest on April 13, and on May 5 it was much stronger and over-scaled in the plot. The last panel of Figure 3 (c) was plotted with the maximum color level 10^1.5 (or 30) times larger than the others. Such a significant day-to-day variation, suggests intermittency of gravity waves.

Figure 4 (a) shows the average of the nightly spectrum at each observation site plotted on the world map. The average phase velocity spectrum at Syowa Station showed a distinct westward propagation with phase speed <50 m/s. At

Shigaraki, AGWs were seen propagating dominantly in east/south-eastward direction that reach phase speed up to ~80 m/s. AGWs at Tomohon showed a preferable propagation direction moving south-eastward with phase speed up to ~100 m/s. It should be noted that the plot of Tomohon was highly affected by a single day, May 5. This again indicates the intermittency of gravity waves, which suggests that the net momentum transportation is highly dependent on the very strong gravity waves with a fewer occurrence frequency.

One can discuss the AGWs propagation by comparing the 2D phase velocity spectrum and the background condition along the propagation path. Figures 4 (b) and (c) show zonal and meridional wind profiles, respectively, from Modern-Era Retrospective Analysis for Research and Applications (MERRA-2; *Gelaro et al.*, 2011) at each observation site. The profiles were averaged over the clear nights of the observed AGWs. We can see that the zonal wind over Syowa Station, which was averaged over 9 clear-sky nights, shows strong eastward wind up to ~50 m/s around ~50 km that could filter out



the waves propagating eastward. Similar conclusion was reported by *Matsuda et al.* [2017] that applied GWs blocking diagram (*Taylor et al.,* 1993; *Tomikawa,* 2015) by using MERRA and MF radar horizontal wind data to discuss the preferred wave propagation direction seen in the phase velocity spectrum. They concluded that the eastward propagating AGWs were restricted to propagate up to the mesopause by critical level filtering.

5       Opposed to the strong zonal wind at Syowa Station, the average zonal wind profile at Shigaraki over 10 clear nights showed relatively weak westward wind, up to ~30 m/s, at around ~50 km altitude. Previous studies on seasonal directionality of AGWs at Shigaraki concluded that east-west propagation anisotropy usually caused by wind filtering of mesospheric jet (e.g. *Nakamura et al., 1999; Ejiri et al.,* 2003; *Takeo et al.,* 2017). However, the westward wind in our current case for April-May 2011 was not more than 30 m/s above stratosphere, and not strong enough for filtering all the gravity waves with

westward propagation out. It is also notable that the average tropopause eastward jet was as strong as 60 m/s. If the gravity wave source is associated with such an eastward jet structure, then the gravity waves generated from there should have horizontal phase velocity distribution shifted to eastward. This could also be the reason of the lack of westward propagating AGW, especially for the ones with higher phase velocities such as over 40 m/s.

       The average zonal wind at Tomohon was pretty small (<20 m/s) which likely had only limited impact on the much

faster AGWs phase speed (~100 m/s). However, we can see clearly in the Figure 4 (a) that the GWs over Tomohon show dominant southeastward propagation. As we already discussed, this distribution was affected by a spectrum of a single day, and it should be highly affected by the location of the strong wave source on that day. *Haffke et al.* [2016] reported that the ITCZ (Intertropical Convergence Zone) is located above the equator between March and April, and started to move north of the equator in May, which could explain the dominant southward propagation of AGWs seen on May 5, 2016 in Figure 3 (c).

The scenario in which the distribution of the source location has more significant effect on the AGWs propagation direction than the background wind in the equatorial region agrees well with what was reported by *Nakamura et al.* [2003]. They reported AGWs characteristics over Tanjungsari, Indonesia (107.9°E, 6.9°S) and by analysing GMS (Geostationary Meteorological Satellite) data, they concluded that the distribution of the tropospheric clouds that were located mainly in the opposite direction of wave propagation played more significant role than the relatively weak background wind. The

meridional wind at each observation site showed weak wind velocity (<5 m/s), which likely had almost no impact on the wave propagation direction.

**4 Conclusion**

       We have developed a user-friendly function based on *Matsuda et al.* (2014) 3D FFT method for airglow data to be

used on IDL. This function can efficiently deal with extensive amounts of imaging data obtained on different years and at various observation sites without bias caused by analysis at different research groups and can treat dynamical/physical effect of AGWs by precisely reflecting amplitude, area and lifetime of each AGW events with reasonable execution time. It can also be applied for airglow data with different observation parameters such as the image sampling interval. This new function has been applied to airglow imaging data at three different latitudes, Syowa Station (69°S, 40°E), Shigaraki





(35°N, 136°E), and Tomohon Observatory (1°N, 122°E) in April – May of selected years. By comparing the 2D phase velocity spectra at each site, we found that the day-to-day variation at Shigaraki was larger than at Syowa Station and the day-to-day variation at Tomohon had even greater variability compared to both, which suggested that the distribution of source region at Shigaraki and Tomohon were likely more variable than at Syowa Station. We also found that the day-to-day variation at Tomohon showed a significant variation on phase spectrum magnitude that could mean intermittency of AGWs. The nightly spectrum average showed a dominant westward wave at Syowa station with phase speed <50 m/s, and at Shigaraki east/south-eastward propagation with phase speed up to ~80 m/s was prevailing. At Tomohon southeastward propagation was found but this was highly affected by a single day when power spectrum was extremely strong. The phase speed at Tomohon reached up to ~100 m/s, which was larger than at the other two sites. Comparison of 2D phase velocity spectra with background wind profiles derived from MERRA-2 data revealed that at Syowa Station, wind filtering played more significant role in filtering out waves propagated eastward. On the other hand, at Shigaraki and Tomohon, the background winds were not strong enough to filter out AGWs propagated in the opposite direction of the observed wave propagation. This shows that the dominant propagation directions seen at Shigaraki and Tomohon Observatory were likely more related to the distribution/characteristic of the source region. Thus, we demonstrated the distinct difference of gravity wave propagation characteristics at three different latitudes by using phase velocity spectrum. These comparison results showed that the usefulness of the phase velocity spectrum to characterize and compare the gravity wave characteristic at different sites, and also usefulness of this new M-transform function on airglow imaging data at these locations. The comparison in the current study was limited to the period of April – May, when we had enough clear night data that were pre-processed. Our future plan is to extend such comparison to include all seasons and months, as well as with more global coverage of data.

**Acknowledgements**

The observation of airglow imaging at Syowa is carried out by JARE (Japanese Antarctic Research Expedition), under MEXT (Ministry of Education, Culture, Sports, Science and Technology). The airglow imager at Tomohon is operated by LAPAN (National Institute of Aeronautics and Space), Indonesia. The airglow imager at Shigaraki is operated by ISEE, Nagoya University in collaboration with the Research Institute for Sustainable Humanosphere, Kyoto University. This research is supported by JSPS KAKENHI grants (JP 15H02137, JP 15H05815 and JP 16H06286), project KP-1 and KP-301 of National Institute of NIPR, and by the JSPS Core-to-Core Program, B. Asia-Africa Science Platforms.





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

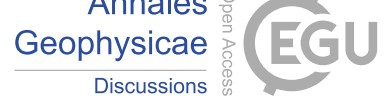



| Program Description | |
|---|---|
| Name | Matsuda_FFT |
| Purpose | Calculate horizontal phase velocity spectral from airglow intensity image data by using 3D FFT. |
| Calling Sequence | Result=Matsuda_FFT(Img) |
| Inputs | Img: Time series of 2D airglow data in geographic coordinate ($x,y,t$) |
| Input Keywords | a. dx, dy, dt: Image resolution in $x$ (m), $y$ (m) and time (s). b. LH_min, LH_max: Minimum and maximum of horizontal wavelength (m) to be processed. c. T_min, T_max: Minimum and maximum of wave period (s) to be processed. d. Vp_min, Vp_max: Minimum and maximum of horizontal phase speed (m/s) to be calculated. e. zpx, zpy, zpt: Dimension of the zero padded image size in $x$, $y$ and $t$ in order to improve the intervals of $k$, $l$, and $\omega$. f. min1, max1: Minimum and maximum of phase velocity spectrum to be plotted. g. Interpolation: Select interpolation method. |
| Outputs | 2D phase velocity spectra ($v_x,v_y$) |
| Remarks | Requires equal sampling interval time resolution ($dt$). |





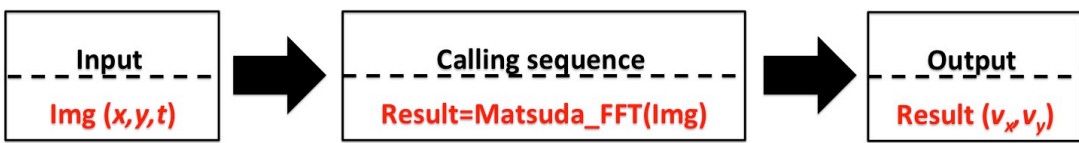

**Figure 1: Flow chart showing the input, how to run the program and the output format.**








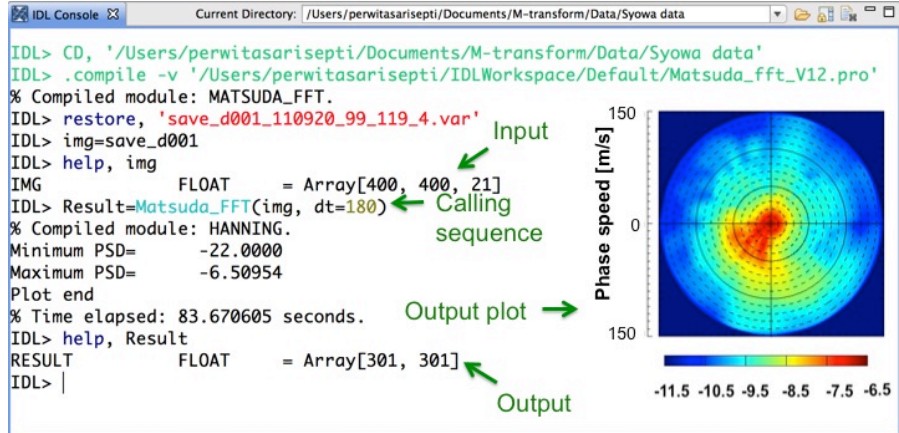

**Figure 2: Example on how to use the M-transform program showing the input, calling sequence, total calculation time and output in 2D phase velocity spectrum.**

20



**Table 2. Summary of airglow imagers in this study**

| Station name (Country) | Syowa (Antarctic) | Shigaraki (Japan) | Tomohon (Indonesia) |
|---|---|---|---|
| Location | 40°E, 69°S | 136°E, 35°N | 122°E, 1°N |
| Institution | National Institute of Polar Research (NIPR) | Nagoya University | National Institute of Aeronautics and Space (LAPAN) |
| Airglow emission | Na (589.6nm) | OI (557.7nm) | OI (557.7nm) |
| Sampling interval ($dt$) (min) | 1 | 5.5 | 2.5 |
| Image Size (km) | 400×400 | 400×400 | 400×400 |
| Minimum wave period ($T\_min$) (min) | 16.5 | 16.5 | 16.5 |
| Maximum wave period ($T\_max$) (min) | 60 | 60 | 60 |



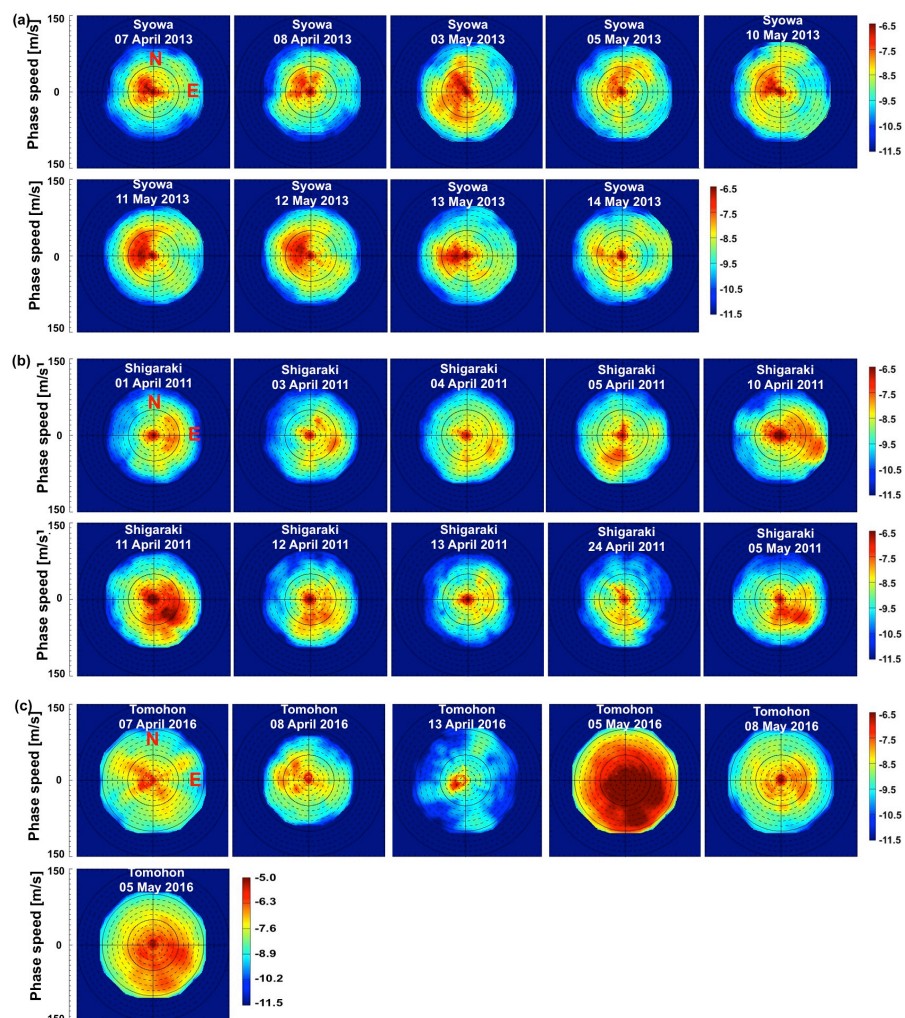

5    **Figure 3: Day-to-day variation of 2D phase velocity spectra between April-May at Syowa Station in 2013 (a), Shigaraki in 2011 (b), and at Tomohon in 2016 (c). The last figure in Figure 3 (c) shows the same figure as 05 May 2016 only with different color bar level to show a better visualization of dominant wave propagation direction.**




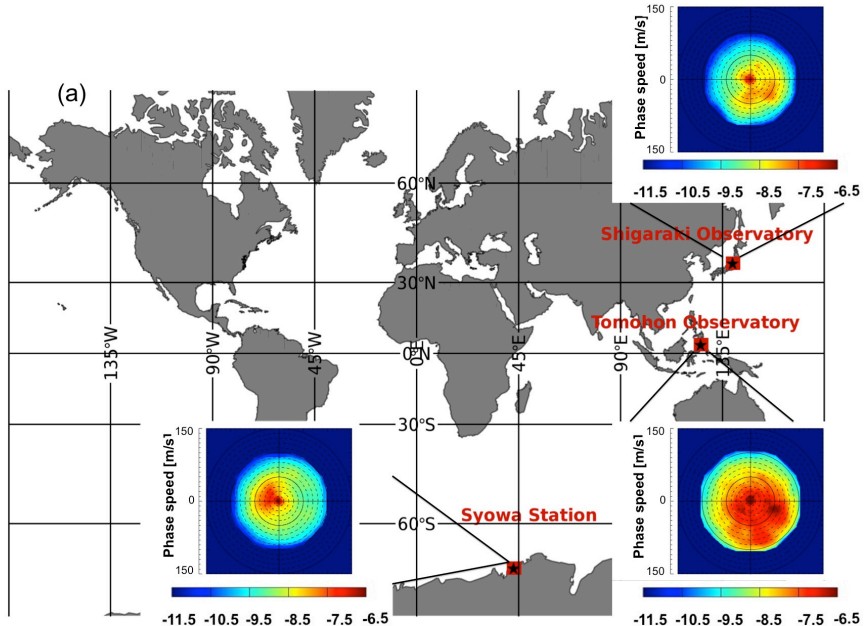

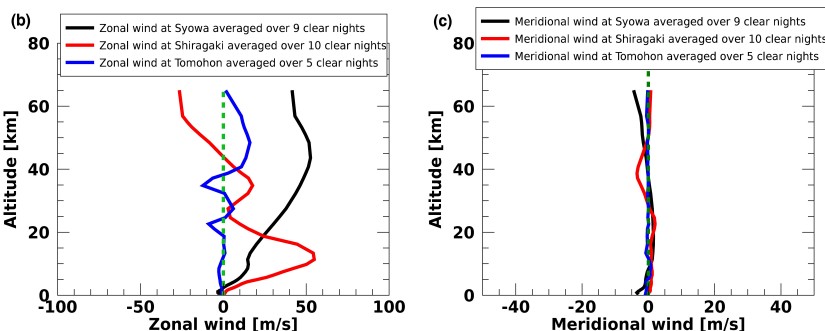

**Figure 4: (a) Average of day-to-day variation of 2D phase velocity spectrum of AGWs seen at Syowa, Shigaraki and Tomohon. (b)**
**Average of zonal wind profiles (positive: eastward) and (c) average of meridional wind profiles (positive: northward) over clear**
**nights of observed GWs from MERRA-2 data.**