# Peer review of "Comparison of gravity wave propagation directions observed by mesospheric airglow imaging at three different latitudes using the M-transform"

_Annales Geophysicae, 2018_

## Referee Comment (RC1) · Anonymous Referee #1 · 9 Oct 2018

The paper describes the IDL package based on Matsuda transform (M-Transform). Airglow images from 3 sites are analyzed using this code to study the gravity wave propagation climatology. Difference among these 3 sites are discussed. The code could be useful for other airglow imager observations in the world. I'd recommend its publication with minor revision.

The English in the paper could be improved with some proofreading by native English speakers. For example:

1. title remove "by"

2. Abstract: line 24: MERRA-2 is less accurate at polar regions because less data going into it.

3. page 2, line 7: "correct" —> "improve"

4. page 2, line 13: "study" — > "studies"

5. page 2 line 22: "airglow" —> "airglow imagers"

---

## Referee Comment (RC2) · Anonymous Referee #2 · 18 Oct 2018

This paper presents a very useful software package for analyzing airglow imagers and provides their phase speed distribution. It is applied to 3 stations and different distributions of phase velocities are found. These are explained with the possible wave source difference and difference in background wind. The writing is clear, and the presentation is concise and to the point. The only area that could be improved is a more detailed analysis of the relationship between wave phase speed distribution and background wind, which may lead to more conclusive findings. This could be another follow-up work.

---

## Author Comment (AC1) · 31 Oct 2018

The authors would like to thank the Anonymous Referee #1 for his/her time and the valuable comments that help to improve the manuscript.

Referee's comment: The English in the paper could be improved with some proofreading by native English speakers.

Response: We have sent our manuscript to professional English editing service to improve the quality of writing.

Referee's comment: (Abstract-line 24) MERRA-2 is less accurate at polar regions because less data going into it.

Response: The MERRA-2 reanalysis is a replacement for MERRA and includes many updates over MERRA (Bosilovich et al., 2015). MERRA-2 assimilates several kinds of satellite data in the polar stratosphere and mesosphere such as GPS-RO, AIRS, Aura/MLS, etc (Fujiwara et al., 2017). Especially, the Aura/MLS temperature data above 5 hPa after 2004 are assimilated to only MERRA-2, which contributes to the significant improvement of its stratospheric and mesospheric representation. In addition, only MERRA and MERRA-2 provide pressure level data above 1 hPa. Our analysis requires wind data at higher levels, therefore we think that the choice of MERRA-2 data in our analysis is suitable.

---

## Author Comment (AC2) · 31 Oct 2018

The authors would like to thank the Anonymous Referee #2 for his/her time and the valuable comments that help to improve the manuscript.

Referee's comment: The only area that could be improved is a more detailed analysis of the relationship between wave phase speed distribution and background wind, which may lead to more conclusive findings. This could be another follow-up work.

Response: Yes, we are preparing a more comprehensive analysis study on the rela-

tionship of phase speed distribution, background wind and AGWs source by including more data for all seasons with more global coverage.

---

## Author Response (AR2)

**Authors' response to Anonymous Referee #1 Comments**

The authors would like to thank the Anonymous Referee #1 for his/her time and the valuable comments that help to improve the manuscript.

The paper describes the IDL package based on Matsuda transform (M-Transform). Airglow images from 3 sites are analyzed using this code to study the gravity wave propagation climatology. Difference among these 3 sites are discussed. The code could be useful for other airglow imager observations in the world. I'd recommend its publication with minor revision.

The English in the paper could be improved with some proofreading by native English speakers. For example:

1.        Title remove "by"

"by" has been removed from the title.

2.        Abstract: line 24: MERRA-2 is less accurate at polar regions because less data going into it.

The MERRA-2 reanalysis is a replacement for MERRA and includes many updates from MERRA (*Bosilovich et al.*, 2015). MERRA-2 assimilates several kinds of satellite data in the polar stratosphere and mesosphere such as GPS-RO, AIRS, Aura/MLS, etc. (*Fujiwara et al.,* 2017). Especially, the Aura/MLS temperature data above 5 hPa after 2004 are assimilated to only MERRA-2, which contributes to the significant improvement of its stratospheric and mesospheric representation. In addition, only MERRA and MERRA-2 provide pressure level data above 1 hPa. Our analysis requires wind data at higher levels, therefore we think that the choice of MERRA-2 data in our analysis is suitable.

3.        page 2, line 7: "correct" —> "improve"
"correct" has been changed to "improve" in P2. Line 7.

4.        page 2, line 13: "study" — > "studies"

"study" has been changed to "studies" in P2. Line 13.

5.     page 2 line 22: "airglow" —> "airglow imagers"
"airglow" has been changed to "airglow imagers" in P2. Line 22.

In addition, to improve the quality of writing we have asked a professional English editing service to check on our manuscript. The changes requested by the Referee #1 are highlighted in blue and the changes suggested by the English editing service are highlighted in yellow in the marked-up manuscript.

**Authors' response to Anonymous Referee #2 comments**

The authors would like to thank the Anonymous Referee #2 for his/her time and the valuable comments that help to improve the manuscript.

This paper presents a very useful software package for analyzing airglow imagers and provides their phase speed distribution. It is applied to 3 stations and different distributions of phase velocities are found. These are explained with the possible wave source difference and difference in background wind. The writing is clear, and the presentation is concise and to the point. The only area that could be improved is a more detailed analysis of the relationship between wave phase speed distribution and background wind, which may lead to more conclusive findings. This could be another follow-up work.

Yes, we are preparing a more comprehensive analysis study on the relationship of phase speed distribution and background wind by including more data for all seasons with more global coverage.

[revised manuscript text omitted]